# 2D-QSAR-guided design of potent carbamate-based inhibitors of acetylcholinesterase

**Meriem Khedraoui[1], El Mehdi Karim[1], Oussama Abchir[1], Abdelkbir Errougui[1], Yasir S. Raouf[2], Abdelouahid Samadi[2]\*, Samir Chtita[1]\***

**1** Laboratory of Analytical and Molecular Chemistry, Faculty of Sciences Ben M'Sik, Hassan II University of Casablanca, Casablanca, Morocco, **2** Department of Chemistry, College of Science, United Arab Emirates University, Al Ain, United Arab Emirates

\* samadi@uaeu.ac.ae (AS); samirchtita@gmail.com (SC)

## Abstract

Alzheimer's disease (AD) causes a progressive decline in memory, along with impairments in other cognitive abilities. The main pharmacological target for Alzheimer's disease (AD) treatment is acetylcholinesterase (AChE), a biochemical enzyme belonging to the cholinesterase (ChE) family. In the search for novel hit compounds with potential as future Alzheimer's therapies, a series of carbamates derivatives were designed and evaluated using computational approaches including QSAR modeling, molecular docking, ADMET profiling, and molecular dynamics simulations. The following study focused on the development of a QSAR model with satisfactory statistical properties. ADMET analysis on the designed ligands, demonstrated good pharmacokinetic properties. Molecular docking identified **M6** as a promising AChE binder with a docking score of -11.200 kcal/mol, while the Donepezil control returned a docking score of -10.800 kcal/mol. The validity of the docked complex was confirmed using molecular dynamics simulations, where the trajectory plots of **M6** were found to be stable and consistent over 100 ns intervals. The enclosed study highlights **M6** as a novel chemical starting point (CSP) (i.e., hit compound) targeting AChE as a potential therapeutic strategy against AD.

## Introduction

Alzheimer's disease (AD) is a complex neurodegenerative condition involving atrophy of healthy brain cells, resulting in progressive dementia and a chronic cognitive decline [1]. This condition results in chronic damage to memory, thinking abilities, and general mental health. In 2024, the World Health Organization (WHO) estimated that there are nearly 10 million new cases of dementia each year worldwide, equating to one new case every 3.2 seconds [2]. In the United States, approximately 6.9 million people aged 65 and older are living with Alzheimer's disease [3] [4]. The lack of effective treatments for this disease results in significant economic, and societal pressure exerting a strong burden on global healthcare systems. [5]. The total healthcare costs for treating AD were estimated at $305 billion in 2020, and as the population ages, these costs are expected to rise to over $1 trillion [6]. What makes it more

**Data availability statement:** All relevant data are within the paper and its Supporting Information files.

**Funding:** AS and YSR acknowledge the support of the UAEU through an UAEU-UOS joint grant (Grant Code G00005017) and an internal Start-up grant 2024 (Grant 12S156), respectively. There was no additional external funding received for this study.

**Competing interests:** The authors have declared that no competing interests exist.

challenging is that science has not yet found an effective treatment for this devastating disease or other diseases causing dementia [7]. Although its underlying pathophysiology remains widely debated (and studied), AD is histopathologically characterized by hyperphosphorylated tau tangles and amyloid plaques in the brain [8]. The human brain contains more than 100 billion neurons and other cells. Nerve cells work together to provide all the necessary connections for functions such as thinking, learning, memory, and planning. The accumulation of these proteins kills these brain cells [9]. Researchers have repeatedly found a deficiency in a brain neurotransmitter (NT), acetylcholine, in Alzheimer's patients. This NT plays a major role in cognitive and logical functions, and is a hydrolysis substrate of cholinesterase enzymes (ChE). In this regard, several efforts have focused on the use of molecular recognition strategies for the upregulation (or maintenance), of acetylcholine levels by targeting AChE and BChE enzymes, or promoting ACh levels in neuronal synaptic clefts [10]. In recent decades, acetylcholinesterase (AChE) has emerged as a potential therapeutic target for AD, as increasing ACh concentrations improve neuronal function [11]. FDA-approved acetylcholinesterase inhibitors include donepezil, rivastigmine, and galantamine, and these marketed drugs are commonly used for the symptomatic treatment of AD [12]. Overall, these small molecules compensate for acetylcholine deficiencies in cholinergic neurons of AD patients, partially alleviating symptoms. However, there remains no scientific evidence that these drugs significantly delay the progression of disease.[13].

The objective of this study was to develop a robust 2D-QSAR model to design carbamate inhibitors with satisfactory predictive activity against AChE using advanced computational drug design techniques. Second, we aimed to predict several pharmacokinetic properties of the top molecules (e.g., absorption, distribution, metabolism, excretion, toxicity). This was followed by molecular docking to predict, quantify, and analyze binding poses to AChE protein. Finally, molecular dynamics (MD) simulations at 100 ns intervals were used to verify the stability of the binding poses.

## Materials and methods

### Calculating molecular descriptors

The QSAR model was developed using a set of previously known chemical compounds comprising 32 molecules which had already demonstrated their biological properties against AChE via in vitro assays [14]. The $IC_{50}$ data was collected from literature, and converted into $pIC_{50}$ values fot the QSAR-2D model development. Molecules were initially subjected to geometry optimization using Gaussian 09 software [15], employing Density Functional Theory (DFT) strategies with the B3LYP/6–31 G basis set. B3LYP is a widely used functional that accurately reproduces the geometry of both small and large molecules [16]. Consequently, several quantum descriptors were also derived, including: Total Energy $E_T$ (eV), Highest Occupied Molecular Orbital Energy $E_{HOMO}$ (eV), Lowest Unoccupied Molecular Orbital Energy $E_{LUMO}$ (eV), and Dipole Moment (eV). We also used Chem3D software to calculate various topological and physicochemical descriptors [17].

### Principal component analysis (PCA)

Before beginning the modeling analysis, the generated descriptors were preprocessed by removing certain descriptors due to the presence of constant descriptors across all the studied molecules, as well as those with missing values. Then, the principal component analysis (PCA) technique was used to carefully examine the remaining molecular descriptors database, to identify molecular descriptors that might be related to the biological activity (e.g., $pIC_{50}$) of the carbamate derivatives. This study provides an opportunity to assess the degree of correlation

between each pair of molecular descriptors by calculating the correlation coefficient (R) and identifying significant intercorrelations based on a correlation threshold ($|R| > 0.95$) [18].

## Model development of QSAR and validation

A crucial step in 2D-QSAR modeling is selecting the most relevant descriptors from all the calculated descriptors. The selection was carried out using the MLR method with the XLSTAT software [19]. And from a final selection of three pertinent molecular descriptors, a total of 43 molecular descriptors were filtered. A linear relationship between $pIC_{50}$ and the descriptors was established by the application of the MLR approach. With distribution ratios of 80% and 20%, respectively, the dataset was split into two sets: a training set of 25 molecules and a test set of 7 molecules.

## 2D-QSAR model validation

Validation tests were conducted to evaluate the explanatory power and predictive ability of the QSAR model [20]. For internal validation, several statistical variables are used to validate the model, such as the coefficient of determination ($R^2$) and the adjusted coefficient of determination ($R^2_{adj}$). Subsequently, the importance of this method is assessed by verifying the cross-validation coefficient of determination ($R^2_{CV}$) using the "leave-one-out" approach. This is a crucial step as a high correlation value indicates better data fitting. According to studies by Golbraikh and Tropsha, it is essential to perform cross-validation, but this alone is not sufficient to demonstrate the predictive abilities of the proposed QSAR model [21]. In addition to internal validation, external validation is necessary; at this stage, the test set was used. The model obtained from the training set is used to analyze the activities of the compounds in the test set, and the coefficient of determination ($R^2_{test\ set}$) is calculated.

## Randomization test

The Y-randomization method is commonly used to assess the robustness of a model. In this method, the values of the dependent variable representing biological activity are randomized, while the independent variable representing the model descriptors remains constant. The randomized data is then used to create a new QSAR model. We performed this using the YRandomization software version 1.2. It is necessary that the squared correlation coefficient of the randomized model ($R^2_{rand}$) is lower than that of the non-randomized model ($R^2$). [22]. This confirms the robustness of the new model. Moreover, it is necessary to examine and highlight the discrepancy between the correlation values of the non-randomized and randomized models.

## Mean effect (ME) and variance inflation factor (VIF)

Calculating the Variance Inflation Factors (VIF) for each of the independent variables in the reported model helps determine the multicollinearity of these variables, and this test was performed using XLSTAT software. The VIF test identifies whether the model's descriptors are related to each other or not. If their estimated VIF values are less than 1, there is no relationship between the descriptors; the model is likely to be accepted if their estimated VIF values are between 1 and 5 [23]; and if their estimated VIF values exceed 10, it indicates the model's instability and necessitates reevaluation [24].

## Applicability domains

The theoretical chemical space of a QSAR model is known as the applicability domain (AD), which encompasses its relevant variables as well as the predicted response [25]. Within the

applicability domain, based on the chemical data used for the model's creation, it is possible to assess the level of uncertainty in recognizing a particular molecule.

The AD is also used to identify outliers in the training set (X-outliers) and to detect molecules that fall outside the AD, in line with the fundamental concept of the standardization method.

Various approaches have been utilized to define the AD of QSAR models. Gramatica described the AD method [26]. Her approach is based on the use of the leverage technique on the dataset. The leverage technique allows for the study of the position of a new molecule within the QSAR model. Thus, the leverage method is employed, as shown in equation (1):

$$hi = X_i^T \left( X^T X \right)^{-1} xi \tag{1}$$

The descriptor matrix of the studied compound is represented by the small descriptor vector **x**. Meanwhile, the large X represents the descriptor matrix, which is generated using the descriptor values from the training set. The warning leverage (h*) was calculated based on equation (2):

$$h^\star = \frac{3 \times (K+1)}{n} \tag{2}$$

The number of training set compounds is n, and the number of model descriptors is K.

## Predictive pharmacokinetic analyses

The initial step in drug discovery requires the prediction of drug-likeness and ADMET properties, as only molecules with satisfactory pharmacokinetic profiles advance to the preclinical stage of drug research [27]. The pharmacokinetic characteristics of the newly designed molecules in this study were predicted using two websites: SwissADME and pkCSM to assess drug-likeness and ADMET properties, respectively [28,29]. The evaluation of the oral bioavailability of the newly designed compounds was based on Lipinski's "Rule of Five" (ROF), a commonly used criterion for assessing oral bioavailability, the rule of five links the physicochemical characteristics of a compound to its biopharmaceutical properties in the human body for oral use [30].

## Molecular docking

The 3D structures of the ligands were subjected to energy minimization using Gaussian 09. The compounds were then converted to PDB format using PyMOL software [31]. Regarding the AChE protein, relevant crystal structures were extracted from the Protein Data Bank (https://www.rcsb.org/), [32]. AChE 4EY7 was selected, with a high resolution of 2.35 Å [33]. After locating the coordinates of the active site: (x = -14.11 Å, y = -43.83 Å, and z = 27.67 Å), The co-crystallized Donepezil ligand was removed in preparation for docking studies., The AChE protein structure was then prepared by removing co-crystallized water molecules using Discovery Studio Visualizer 2016 [34]. Using AutoDock version 1.5.7, polar hydrogen atoms and Gasteiger and Kollman charges were added, and missing residues were repaired to generate a suitable (and physiologically relevant) protein structure duringreceptor-ligand dockings and MD simulation [35]. Molecular docking study parameters were set to default values, and a grid box of size 40 × 40 × 40 Å centered on the active site with a grid spacing of 0.375 Å was generated. Ligands were then docked into the active site of AChE. as flexible molecules, allowing rotatable bonds, while the AChE protein was treated as rigid [36]. Resulting binding poses were then evaluated using Discovery Studio 2016.

## Molecular dynamics

A 100 ns molecular dynamics (MD) simulation was performed to analyze the structural stability of the compound developed in this study using GROMACS 2020.4 software [37]. The protein topology was generated using the OPLS-A force field, and the SwissParam server was used to generate the ligand topology [38]. These topology files were merged. The protein-ligand system was then placed in a cubic box consisting of a TIP3P water model. To neutralize the system's charge, $Na^+$ and $Cl^-$ ions were incorporated. Subsequently, the steepest descent algorithm was used to minimize the entire system. After that, a 100 ns simulation was carried out under NVT and NPT ensembles at a constant temperature (300 K) and pressure (1 atm). The trajectories were integrated every two femtoseconds (fs), and the atomic coordinates of the simulated structures were recorded every 10 ps. Throughout the trajectory, the root mean square deviation (RMSD), root mean square fluctuation (RMSF), radius of gyration (Rg), number of hydrogen bonds, and solvent-accessible surface area (SASA) were studied.

## Results and discussion

Different QSAR models were developed with a high coefficient of determination; however, a more robust, efficient, and reliable model was selected as the best model due to the significance of its parameters. Indeed, it shows the highest values of $R^2 = 0.81$, $R^2_{adj} = 0.78$, and $Q^2_{cv} = 0.56$. The robustness and accuracy of the QSAR model were assessed using statistical parameters. The QSAR model is presented below.

$$pIC_{50} = -1.454 + 0.005 * \textbf{Connolly Accessible Area} - 54.459 * E_{LUMO} + 0.285 * \textbf{Hydrogen\%} \quad (3)$$

The external validation of the model allowed us to predict the activity of an external set, resulting in a test set regression coefficient $R^2_{Test\,set} = 0.82$. These results clearly demonstrate the external validity of the model. A group of data points around the reference line showing predicted activity versus experimental activity, as illustrated in Fig 1, indicates the robustness and reliability of the selected QSAR model. Our MLR model was chosen as the most effective because it exhibited statistical parameters similar to those reported for a robust model [39, 40]. The three descriptors have VIF values ranging from 1 to 5 in Table 1, suggesting that the developed QSAR model was statistically significant and thus stable and acceptable. The absolute t-statistic value of each descriptor is greater than 2, indicating that the chosen descriptors were significant [41]. Additionally, the p-values obtained for these descriptors in the model at a 95% confidence level were less than 0.05. This requires accepting the alternative hypothesis that there is a direct relationship between the biological activity of each compound and the descriptor that influences the constructed model. Consequently, the null hypothesis is rejected, which states that there is no direct link between the biological activity of each compound and the descriptor that influences the constructed model. The results of the randomization Y test showed very low $R^2_{rand} = 0.27$ and $Q^2 = -1.01$ values, suggesting that the developed model is stable, solid, and reliable. When the randomization-Y coefficient $cR^2p = 0.76$ is greater than 0.50, it indicates that the developed model is powerful and not dependent on chance.

## Applicability domain

QSAR models use experimental data that are restricted to a group of molecules, making each extracted model unique. In other words, the model is only valid for molecules similar to the input data. The applicability domain of each model can be identified based on the range of applicability. The region where the predicted biological activity is specified for each model

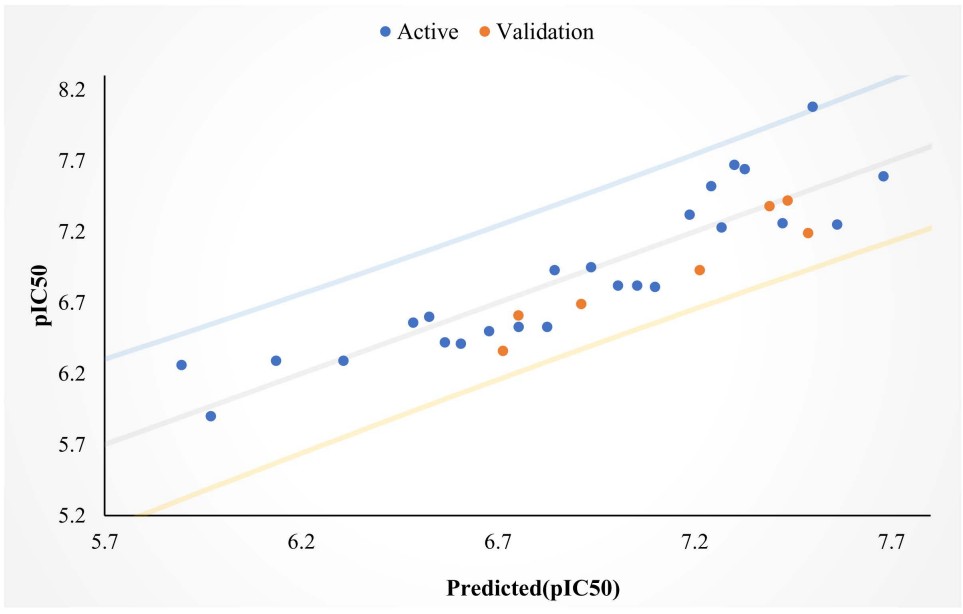

**Fig. 1. Correlation between expected and observed biological activity values.**

**Table 1. Accepted QSAR model validation tools: evaluation of predictive performance and robustness [42].**

| Validation tools | Interpretation | Acceptable value | Developed model value and remarks |
|---|---|---|---|
| $R^2$ | Co-efficient of determination | ≥0.6 | 0.81<br>Pass |
| $Q^2_{cv}$ | Cross-Validation Co-efficient | >0.5 | 0.74<br>Pass |
| $P_{(95\%)}$ | Confidence interval at 95% confidence level | <0.05 | 0.0001-0.009<br>Pass |
| $R^2_{Test\ set}$ | Co-efficient of determination of external and test set | ≥0.5 | 0.82<br>Pass |
| $R^2_{adj}$ | Adjusted R-squared | >0.6 | 0.78<br>Pass |
| $cR^2p$ | Coefficient of determination for Y-randomization | >0.5 | 0.76<br>Pass |
| $t_{test}$ | t-Statistic value | >2 | 2.86-4.90<br>Pass |
| VIF | Variance Inflation Factor | <10 | 1.80-2.82<br>Pass |

[43]. The Williams plot (Fig 2) was used to visually represent the applicability domain of the model developed in this study. The normal control values for outlier Y values (cross-validated residuals) were established as ±3σ. The data set includes 32 compounds, with 7 serving as a test set and the remainder as a training set. This means that the warning leverage has a value of 0.48. All observations fall within the applicability domain (three units of deviation and the warning leverage h*) as shown in Fig 2, which indicates the reliability of the extracted model for predicting the activity of new compounds.

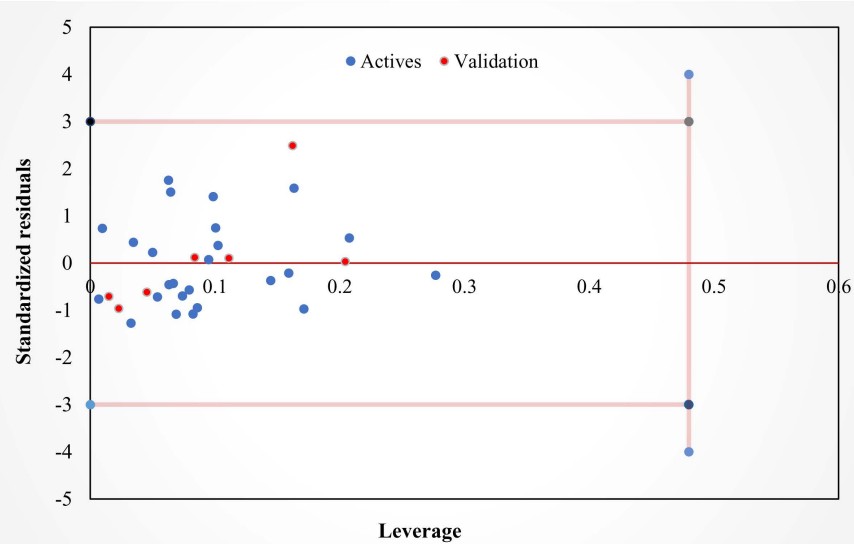

**Fig. 2. Applicability domain of the developed QSAR model.**

## Molecular design based on ligands

To create more effective carbamate analogues as AChE inhibitors, an in-silico screening approach was implemented based on the QSAR model. The template molecule used for the design was molecule 8 (S1 Table), which was chosen from the training set due to its high inhibitory capacity (pIC$_{50}$ = 8.08). According to the suggested model (Eq. 1), the main features influencing antiacetylcholinesterase activity are Connolly Accessible Area, E$_{LUMO}$, and Hydrogen%. The t-test value of each descriptor was calculated to assess their impact level (Table 1). Indeed, the variance in antiacetylcholinesterase activity is strongly influenced by the descriptor with the highest absolute t-test value. Antiacetylcholinesterase activity is significantly influenced by Connolly Accessible Area and Hydrogen%. To create new substances with improved antiacetylcholinesterase activity, the structure of the compound was modified by adding methyl and hydroxyl groups to the model molecule. The predicted MLR model was used to predict the AChE inhibitory activity of seven hypothetical designed compounds listed in S2 Table. All their activities were more potent than the template molecule, with a predicted pIC$_{50}$ ranging from 8.20 to 8.76.

## ADMET properties and drug likeness

According to Lipinski's Rule of 5 (Ro5 for oral bioavailable small molecules, a compound is considered more "drug-like" if it has a MW <500, HBD < 5, HBA < 10, and a LogP < 5. [44]. These are summarized in Table 2 which evaluates molecular weight, the number of hydrogen bond donors, hydrogen bond acceptors, and the lipophilicity partition co-efficient (logP). As shown by the results, none of the designed compounds violat Lipinski's Rule of Five (RO5). This suggests that if successful during synthesis and development, these compounds could be orally bioavailable. [45].

The likelihood of therapeutic success for drug-like molecules, is also determined by their pharmacokinetic properties. The Caco-2 cell line, which consists of human colorectal adenocarcinoma epithelial cells, is frequently used as an in vitro model of the human intestinal mucosa to predict oral drug absorption. This prediction is performed by studying the logarithm of the apparent permeability coefficient (log Papp; log cm/s). If the predictive log Papp

**Table 2. Drug-likeness parameters according to Lipinski.**

|  | Molecular weight (g/mol) | Donors HBD | Acceptors HBA | Calculated LogP |
|---|---|---|---|---|
| M2 | 614.068 | 0 | 4 | 9.381 |
| M3 | 545.167 | 0 | 4 | 8.600 |
| M4 | 545.167 | 0 | 4 | 8.600 |
| M5 | 549.199 | 0 | 4 | 9.048 |
| M6 | 555.247 | 0 | 4 | 8.872 |
| M7 | 689.050 | 0 | 3 | 11.200 |

values exceed 0.90 cm/s, a compound is considered to have high permeability through Caco-2 cells [46]. According to Table 3, the compounds studied have Caco-2 permeability (log Papp) ranging from 0.8 to 0.9 cm/s. Compounds M6 and M7 have a log Papp greater than 0.9 cm/s, suggesting high permeability through Caco-2. Compounds M2, M3, M4, and M5, which have a log Papp less than 0.9 cm/s, are expected to exhibit low permeability through Caco-2.

Microvascular endothelial cells of the central nervous system (CNS) form the blood-brain barrier (BBB), a unique biological barrier. Everything entering and leaving the brain is filtered by these cells. CNS homeostasis is primarily maintained by the BBB, which limits the transport of toxic substances and removes metabolites from the brain [47]. The LogBB value, which is the logarithm of the ratio between drug concentration in the brain and concentration in the blood measured at equilibrium, indicates how easily a compound crosses the BBB. Compounds with a logBB greater than 0.3 are those that easily pass through the BBB, while those with a logBB less than -1.0 are more difficult to cross. According to the results, all the designed molecules are distinguished by their ability to cross the blood-brain barrier and achieve bioavailability in neurological pathways.

The most common metabolism mode of small molecule drugs is oxidative metabolism by cytochrome P450 enzymes (CYP450) [48]. Of these enzymes, CYP2D6 and CYP3A4 play crucial roles in drug metabolism, with CYP3A4 cited as being the primary metabolizing enzymes for almost half of all drugs. Therefore, predicting CYP inhibition is crucial in drug development [49]. The molecules were identified as non-substrates and inhibitors of the CYP2D6 microsomal enzyme, while all the compounds are metabolized by CYP3A4 as substrates and not inhibitors, implying that these molecules will not disrupt the biotransformation of drugs metabolized by these enzymes [50]. Drug duration in the body is determined by clearance, a parameter that represents the correlation between drug concentration and elimination rate [51]. Consequently, the recent compounds have high clearance values, ensuring optimal drug retention in the body. Toxicity levels must be considered when searching for drugs. The

**Table 3. ADMET properties for the selected compounds.**

|  | Caco2 | HIA% | BBB | CYP3A4 substrate | CYP2D6 substrate | CYP2D6 inhibitior | CYP3A4 inhibitior | Total Clearance | AMES toxicity | Hepatotxicity | SS |
|---|---|---|---|---|---|---|---|---|---|---|---|
| M2 | 0.809 | 88.865 | −0.409 | Yes | No | Yes | No | 1.316 | No | No | No |
| M3 | 0.838 | 91.125 | −0.340 | Yes | No | Yes | No | 1.125 | No | No | No |
| M4 | 0.838 | 91.125 | −0.340 | Yes | No | Yes | No | 1.158 | No | No | No |
| M5 | 0.813 | 90.113 | −0.393 | Yes | No | Yes | No | 1.453 | No | No | No |
| M6 | 0.940 | 91.565 | −0.390 | Yes | No | Yes | No | 1.510 | No | No | No |
| M7 | 0.925 | 84.896 | 0.955 | Yes | No | Yes | No | 1.692 | No | No | No |

**HIA%:** Intestinal absorption (human), **BBB**: permeability to brain blood barrier.

SS Skin Sensitisation

predictive toxicity results of the compounds are presented in Table 3. The Ames test predicts whether a compound has mutagenic, and thus carcinogenic, properties. In reality, none of the molecules are carcinogenic, hepatotoxic, or cause skin sensitization.

## Molecular docking

One commonly employed method for discovering new inhibitors is molecular docking. This method has allowed for the identification of potent compounds that may resemble the co-crystallized structure or be used as new leads. In this study, LGA was used, treating ligands as flexible entities. The research was conducted using AutoDock Vina 4.2 [52], which employs a semi-empirical free energy force field to calculate scores (Eq. 4). The energies involved in the protein-ligand binding, such as van der Waals energies, electrostatic energies, hydrogen bond energies, desolvation energies, and torsion penalties, play a role in the scoring function.

$$V = W_{vdw} \sum_{i,j} \frac{A_{ij}}{r_{ij}^{12}} + \frac{B_{ij}}{r_{ij}^{6}} + W_{hbond} \sum_{i;j} E(t) \left\{ \frac{C_{ij}}{r_{ij}^{12}} + \frac{D_{ij}}{r_{ij}^{10}} \right\} + W_{ele} \sum_{i,j} \frac{q_i q_i}{e(r_{ij})r_{ij}} +$$

$$W_{sol} \sum_{i,j} (S_i V_j + S_j V_i) e^{\left( -\frac{r_{ij}^2}{2\sigma^2} \right)} + W_{tor} N_{tor}. \tag{4}$$

The re-docking method was used to assess the efficiency and accuracy of the molecular docking algorithms by superimposing the co-crystallized ligand onto the docked ligand. The superposition results shown in Fig 3 reveal a root-mean-square deviation (RMSD) of 0.28 Å, which is less than 2 Å, demonstrating the accuracy and precision of the predicted pose. Thus, the docking verification process was successfully studied.

Two compounds designed on the initial pharmacophore of a carbamate, selected after the ADMET test, were evaluated for their ability to bind to the AChE protein (PDB ID: 4EY7). The results of this docking study are represented in Fig 4. The compounds studied, namely compound M6 and compound M7, showed interesting docking scores of -11.2 and -10.9 kcal/mol, respectively. These scores were higher than that of the co-crystallized ligand, which was -10.8 kcal/mol. Illustration 4 shows how these compounds were integrated into the active site of the AChE protein by interacting with amino acid residues.

Compound M6 demonstrated the highest score, with a binding energy value of -11.2 kcal/mol relative to the active site. It formed multiple interactions with the active amino acid residues in the AChE binding site (Fig 4a). It established three hydrogen bonds with the amino

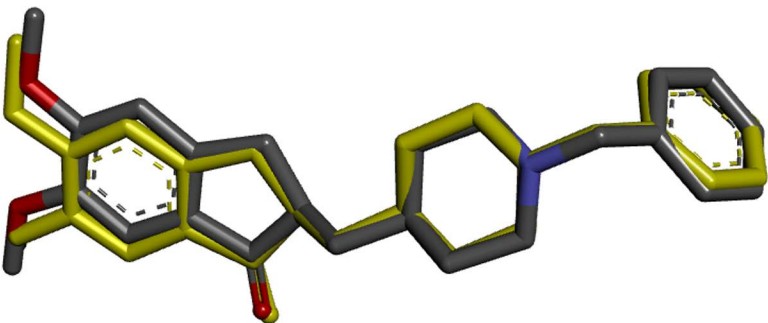

**Fig. 3. Assessment of molecular docking accuracy by superimposing the docked ligand and the co-crystallized ligand.**

acids PHE295, ARG296, and GLY121, as well as sixteen hydrophobic interactions with residues TYR341, TRP286, TRP86, TYR124, LEU76, TYR337, PHE338, and HIS447.

On the other hand, compound M7 exhibited a slightly lower free energy of -10.9 kcal/mol compared to compound M6. Fig 4b shows that it interacts with the AChE protein through hydrophobic interactions with residues TRP286, LEU289, TRP86, TYR337, PHE338, TYR341, and HIS447.

## Molecular dynamics simulation studies

Molecular dynamics (MD) is used in drug discovery, and structure-based drug design to study the movement and trajectory of molecules in the presence of other molecules, proteins, receptors while also studying potential intermolecular interactions within the two-body system. It helps in understanding conformational changes of molecules, structural characteristics of proteins, and drug-protein interactions. In this study, the AChE-M6 complex was examined through MD simulations. A detailed analysis of the RMSD, RMSF, SASA, radius of gyration, and HBond trajectories was conducted.

### RMSD analysis

RMSD (Root Mean Square Deviation) reflects the average displacement variations in the position of an atom for a specific frame relative to a reference frame. RMSD simulation provides data on the structure of the protein and ligand, as well as the stability and equilibrium of the system [53, 54]. As the RMSD variations of the protein increase, it becomes more unstable, and vice versa [55]. The average RMSD of the unbound AChE protein ranged between 15 and 17 Å and remained stable throughout the MD simulation period. The AChE protein bound to ligand M6 maintained an average RMSD of 3 Å throughout the MD simulation period. Regarding the ligand, Fig 5 shows an increase in the ligand RMSD to 6 Å at 20 ns, after experiencing fluctuations in RMSD between 5 and 7 Å. It then stabilized at 6 Å around 77 ns until the end of the simulation (Fig 5).

### Root mean square fluctuations (RMSF)

To evaluate the dynamic behavior of residues, it is necessary to measure the root mean square fluctuations (RMSF) of the Ca atoms of all residues in both the apo protein and the AChE_M6 complex systems. RMSF provides valuable data on structural flexibility and variations in different parts of the protein [56, 57]. Fluctuations are generally larger when residues are unstable, while lower RMSF values indicate residue stability. Fig 6 shows the RMSF of the Ca atoms for the apo and AChE-M6 complex.

Most residues in the AChE protein, whether uncomplexed or complexed, had RMSF values less than 2 Å, suggesting that the residues were stable during the MD simulation. Additionally, the fluctuation curves for the complex were similar to those of the apo form, further confirming the stability of the AChE-M6 complex. However, significant conformational changes were observed in the N-terminal and C-terminal residues. This can be explained by the fixed position of the terminal residues, which tend to vary.

Nominal RMSF fluctuations can be attributed to the dynamics of the ligands within the binding pocket. The average RMSF values for each M6 atom were also calculated, as illustrated in Fig 7. Some variations were observed in their RMSF values, indicating dynamic displacement from their initial positions.

## Solvent-accessible surface area (SASA)

According to the system conditions, solvent behavior varies, making solvent-accessible surface area (SASA) a useful method for explaining protein conformational dynamics.

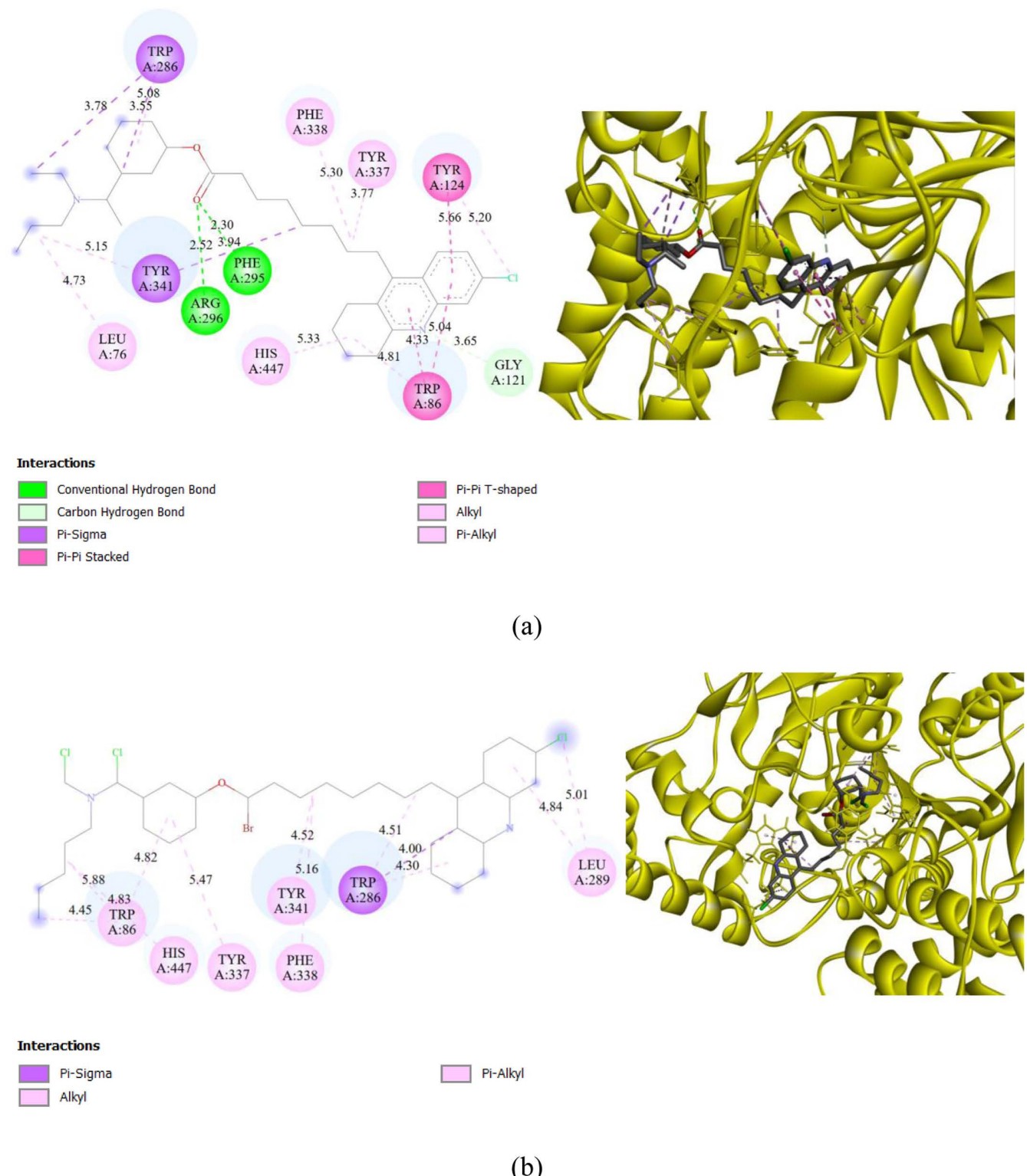

**Fig. 4. Two-dimensional interactions between AChE and the ligands (a): M6 and (b): M7.**

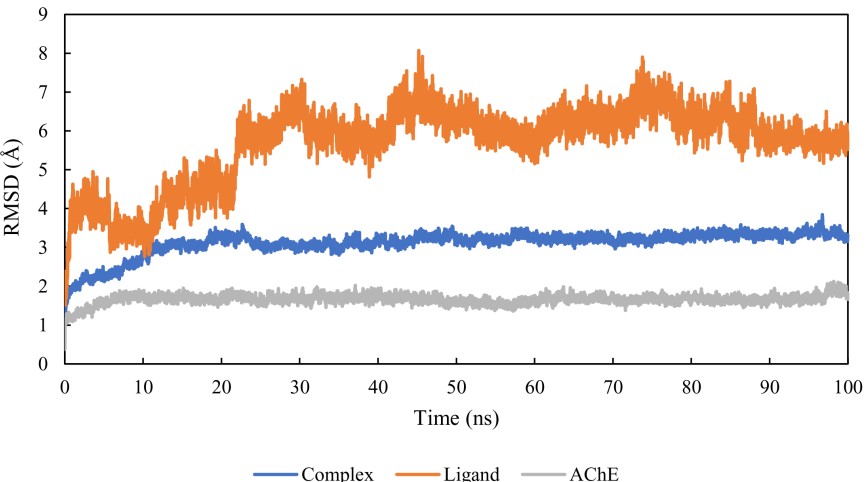

**Fig. 5. RMSD of AChE, M6, and AChE-M6.**

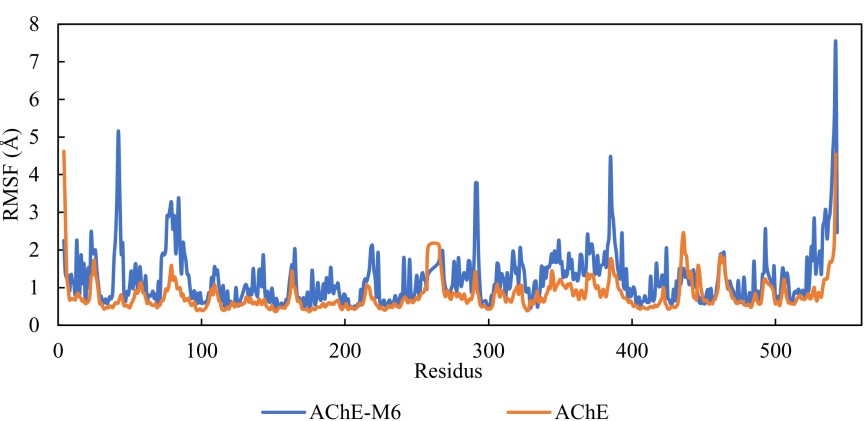

**Fig. 6. RMSF of AChE and AChE-M6.**

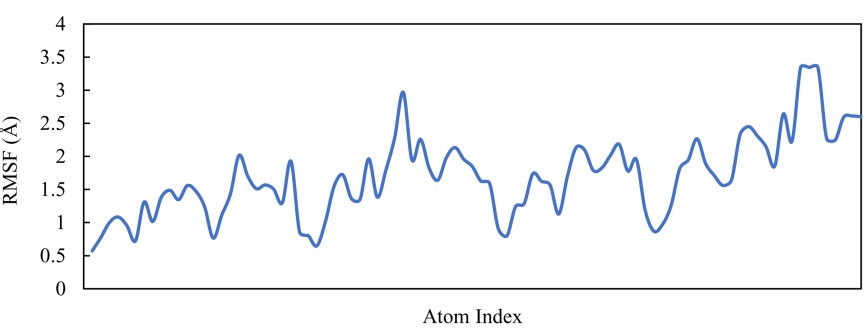

**Fig. 7. R MSF of ligand M6.**

Time-dependent SASA values were calculated for both the apo and the docked complex using a 100 ns MD simulation. According to Fig 8, it can be observed that the SASA of the AChE-M6 complex is shifted and its value decreases. The SASA values are lower in the M6 complex system compared to the apo system, suggesting that the SASA has reached equilibrium throughout the simulation, indicating the structural stability of the complex (Fig 8).

### Radius of gyration

The radius of gyration (Rg) plays a crucial role in directly relating to the volume of the tertiary structure and the overall conformational structure of a protein [58, 59]. It provides essential information about a protein's stability, with a higher Rg suggesting a looser packing. The average Rg values for both the apo and AChE-M6 complex systems were calculated. Their Rg values were 23.3 and 23 Å, respectively. The Rg results highlight the stability of the complex's structure during the simulation (Fig 9).

### Hydrogen bond dynamics

The stability of proteins primarily relies on intramolecular hydrogen bonding [60]. The analysis of intramolecular hydrogen interactions can provide valuable data on the overall stability of protein structures. Furthermore, studying hydrogen interactions between molecules allows us to examine the polar interactions between a protein and a ligand, revealing the directionality and specificity of these interactions, which is a crucial element of molecular recognition [61, 62]. We investigated the dynamics of hydrogen bonds formed in the AChE-M6 complex during the simulation to confirm and assess the stability of the complex. Specifically, hydrogen bonds were formed throughout the simulations. It was observed that the average number of hydrogen bonds was two, indicating that the M6 compound remains in the binding pocket throughout the 100 ns (Fig 10).

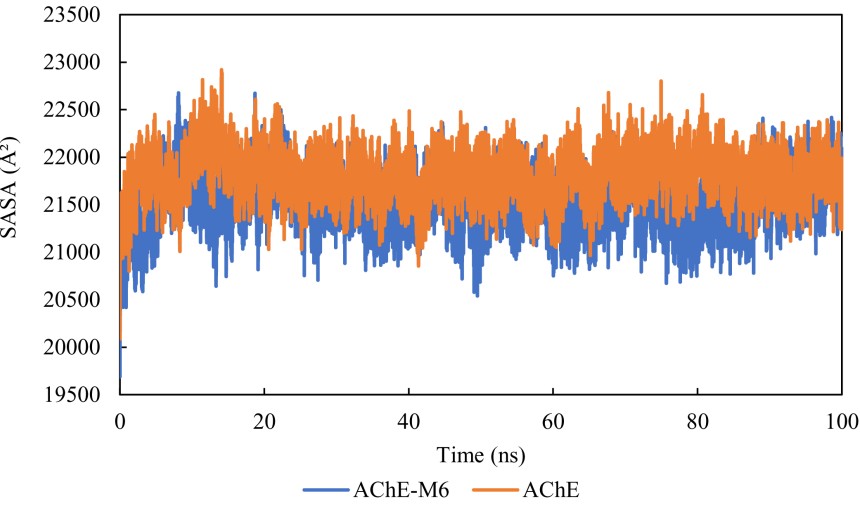

**Fig. 8. Solvent-accessible surface area (SASA) of AChE-M6 complex and the AChE protein.**

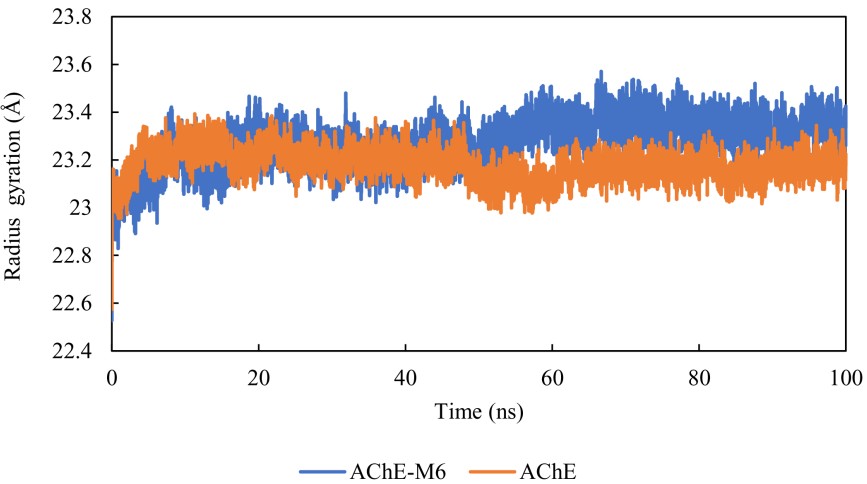

**Fig. 9. Radius of gyration of the AChE protein and AChE-M6 complex.**

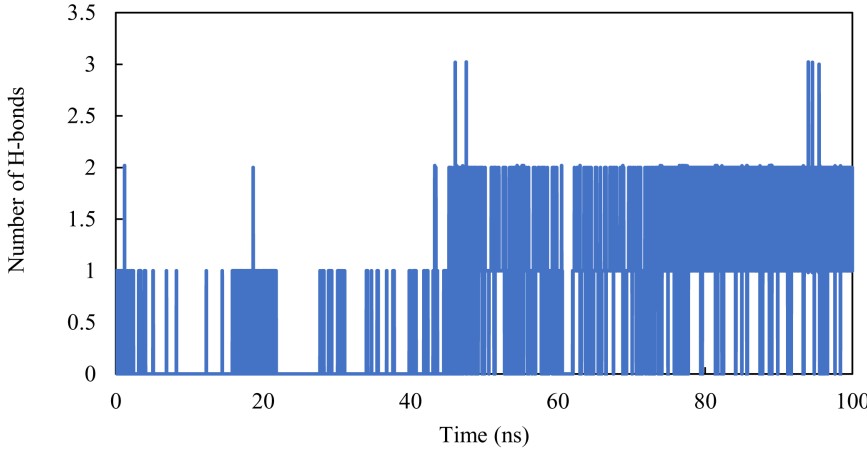

**Fig. 10. Number of hydrogen bonds in the AChE-M6 complex.**

## Conclusion

In this work, a QSAR study is presented for a series of 32 carbamate derivatives to determine their activity against the target protein AChE. The QSAR model was developed using various statistical parameters. Internal and external validation confirmed the reliability and robustness of the model, demonstrating its predictive power as well as statistical significance. Randomization testing, variance inflation testing, and applicability domain assessment were conducted to ensure the robustness of the developed MLR model. Three molecular descriptors (Connolly accessible area, H %, $E_{LUMO}$) showed a higher correlation with biological activity. In this research, new molecules were developed based on the structural changes of one of the most active carbamate derivatives. These were subjected to ADMET prediction studies, molecular docking, and molecular dynamics simulations. The results indicate that the M6 and M7 molecules meet all ADMET criteria and exhibit drug-like properties. Molecular docking studies show

relevant interaction with the AChE protein. The designed M6 compound was tested through molecular dynamics simulation, which demonstrated its stability and cohesion within the active site over 100 ns. This suggests that the proposed M6 molecule could be a potential candidate for further studies in the field of drug development against Alzheimer's disease.

## Supporting information

**Table S1. Structures and pIC$_{50}$ values of the 32 studied compounds.**
(DOCX)

**Table S2. Structures and predicted pIC$_{50}$ values of new designed compounds.**
(DOCX)

## Author contributions

**Conceptualization:** Meriem Khedraoui, El Mehdi Karim, Oussama Abchir.

**Data curation:** Meriem Khedraoui, El Mehdi Karim.

**Formal analysis:** Meriem Khedraoui, El Mehdi Karim.

**Funding acquisition:** Yasir S. Raouf, Abdelouahid Samadi.

**Investigation:** Samir Chtita.

**Methodology:** Samir Chtita.

**Project administration:** Samir Chtita.

**Resources:** Samir Chtita.

**Software:** Samir Chtita.

**Supervision:** Samir Chtita.

**Validation:** Yasir S. Raouf, Abdelouahid Samadi, Samir Chtita.

**Visualization:** Oussama Abchir, Abdelkbir Errougui, Yasir S. Raouf, Abdelouahid Samadi, Samir Chtita.

**Writing – original draft:** Meriem Khedraoui, El Mehdi Karim, Oussama Abchir.

**Writing – review & editing:** Meriem Khedraoui.

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
