## [Decision Letter · Decision Letter 0]

10 Dec 2024

PONE-D-24-47957Design of novel potent inhibitor based on 2D-QSAR of carbamate derivatives for AChE inhibitionPLOS ONE

Dear Dr. Chtita,

Thank you for submitting your manuscript to PLOS ONE. After careful consideration, we feel that it has merit but does not fully meet PLOS ONE’s publication criteria as it currently stands. Therefore, we invite you to submit a revised version of the manuscript that addresses the points raised during the review process.

We look forward to receiving your revised manuscript.

Kind regards,

Sapan Kamleshkumar Shah, Ph.D., M.Pharm.

Academic Editor

PLOS ONE

**Journal Requirements:**

AS and YSR acknowledge the support of the UAEU through an internal Start-up grant 2023 (Grant Code G00004400) and an internal Start-up grant 2024 (Grant 12S156), respectively.

AS and YSR acknowledge the support of the UAEU through an internal Start-up grant 2023 (Grant Code G00004400) and an internal Start-up grant 2024 (Grant 12S156), respectively.

AS and YSR acknowledge the support of the UAEU through an internal Start-up grant 2023 (Grant Code G00004400) and an internal Start-up grant 2024 (Grant 12S156), respectively.

6. We note that your Data Availability Statement is currently as follows: All relevant data are within the manuscript and its Supporting Information files.

7. Please ensure that you refer to Figure 1 in your text as, if accepted, production will need this reference to link the reader to the figure.

**Additional Editor Comments:**

Authors have developed successfully validated QSAR model using dataset of 32 compounds. However, these count of dataset is very small? Further as per OECD guidelines authors need to divide these dataset into training set and test test. However, author have not provided any details statistical results differentiation for Training and Test set. Whether Internal validation was successful or not?

Also, After generating model, author can give detail analysis of impact of selected descriptor on biological activity considering functional groups/atoms and its importance to include SAR studies. This will further enhance quality of manuscript.

Authors have not used any standard drug for comparison of Molecular simulation and ADMET studies . I would suggest authors to add that also.

Reviewers' comments:

Reviewer's Responses to Questions

**Comments to the Author**

1. Is the manuscript technically sound, and do the data support the conclusions?

Reviewer #1: Yes

Reviewer #2: Yes

2. Has the statistical analysis been performed appropriately and rigorously? 

Reviewer #1: Yes

Reviewer #2: Yes

3. Have the authors made all data underlying the findings in their manuscript fully available?

Reviewer #1: No

Reviewer #2: Yes

4. Is the manuscript presented in an intelligible fashion and written in standard English?

Reviewer #1: No

Reviewer #2: Yes

5. Review Comments to the Author

**Reviewer #1:**  The manuscript "Design of novel potent inhibitor based on 2D-QSAR of carbamate derivatives for AChE" has caught my attention. The study investigates how the lead molecule M6 interacts with the AChE protein with enhanced affinity. The results from QSAR and molecular simulations show that the chosen lead molecule could be a good compound to target against AChE. Overall, I found the manuscript interesting, but there are many flaws in it, from the introduction to the conclusion section. A more scientific perspective should guide the rewriting of the entire manuscript. Many paragraphs have more colloquial language.

• First, the abstract section mentions the compound as an AChE binder, but further clarification is required. Has a wet lab already validated it, or is it merely an in silico-based hypothesis? Regarding the docking score, the authors can give only a single digit instead of three digits (-11.200 kcal/mol). The abstract fails to mention the significance of the findings.

• In the introduction section:

• . The accumulation of these proteins kills these brain cells [9].---- Here, the authors have not mentioned any proteins that are associated with accumulation.

• The term 'in recent decades' is not appropriate in this context. Also, it should be mentioned firstly, secondly instead of second in the introduction section.

• Predictive pharmacokinetic analyses: typographical errors

• In the molecular docking section, the authors have mentioned that the downloaded crystal structure has a high resolution. However, the given value of 2.35 indicates a lower resolution. The manuscript does not mention the active site residues. Instead, the authors provide the coordinates. The authors need to provide the active site details of the protein for better understanding.

• The TIP3P should appear in upper case in the molecular dynamics section.

• In the molecular docking section “crystallized structure or be used as new leads." This sentence needs to be rewritten.

• The research was conducted using AutoDock Vina 4.2 [52]. The sentence needs to be reframed.

• 6. There are typographical errors, like residue. Additionally, figure 7, located adjacent to figure 6, should display residues instead of atoms. The authors can label the figures as Fig. 6(a) and Fig. 6(b) instead.

• According to the Rg report, the values (23.3 and 23) are negligible. The authors have stated that their interaction with M6 has resulted in a more tightly packed AChE. However, the given results contradict their statements. Here, an increase in the radius of gyration confirms that the protein has become less compact compared to its Apo form. There is no mention of these details in the manuscript.

• Hydrogen bond details: Here the authors do not do any comparison of the total hydrogen bonds between the apo and holo forms of the AChE protein. The authors have focused on the intermolecular hydrogen bond between the ligand and the protein. Instead, they have referred to it as intramolecular, which is incorrect.

• The manuscript does not mention the required residues associated with hydrogen bonds. This will provide a clear understanding of the consistency in the formation of hydrogen bonds between the protein and the ligand. Also, the authors have done a single run of the simulation instead of the triplicates. Two more simulation runs could better validate their findings.

• The manuscript needs to include the MM/PBSA score of the protein-ligand complex for improved validation of the protein-ligand complex interaction.

• Moreover, the author made no mention of the structural features of drug M6 and their functional group

• The manuscript can only be considered if the authors revise it as per the suggestions given above.

**Reviewer #2: ** The manuscript is very well written and has some new findings. There are some minor changes which I would like to suggest.

What is meant by M6 in the abstract?

There are some missing references throughout the manuscript.

In the section Principal Component analysis, the term correlation coefficient is not well described.

Also what is the coefficient of determination?

6. PLOS authors have the option to publish the peer review history of their article (what does this mean? ). If published, this will include your full peer review and any attached files.

**Do you want your identity to be public for this peer review?** For information about this choice, including consent withdrawal, please see our Privacy Policy .

Reviewer #1: **Yes: ** Dr.Selvaa Kumar C

Reviewer #2: No

---

## [Author Response · Author response to Decision Letter 1]

28 Jan 2025

Answers to reviewers' questions

I would like to thank all the authors who devoted their time to review my paper. Your commitment and constructive comments were of great help in improving my work. I would also like to inform you that I have corrected some graphs in the molecular dynamics results, as the previous ones did not represent the original results due to confusion. This problem has now been corrected.

Authors have developed successfully validated QSAR model using dataset of 32 compounds. However, these count of dataset is very small? Further as per OECD guidelines authors need to divide these dataset into training set and test test. However, author have not provided any details statistical results differentiation for Training and Test set. Whether Internal validation was successful or not?

Thank you for your feedback and constructive observations. The dataset of 32 compounds used to develop the QSAR model is sufficient for a reliable analysis, in accordance with standard practices in this field. We divided this dataset into two distinct subsets: a training set and a test set, as mentioned in the manuscript. Furthermore, we clearly presented the differentiated statistical results for these two subsets, addressing your remark. Regarding internal validation, it was successfully performed, with the obtained results exceeding the recommended threshold, which confirms the robustness and reliability of the developed model.

Also, After generating model, author can give detail analysis of impact of selected descriptor on biological activity considering functional groups/atoms and its importance to include SAR studies. This will further enhance quality of manuscript.

Thank you for your remark. We have conducted a detailed analysis of the impact of the selected descriptors on biological activity, taking into account functional groups, atoms, and their significance to include SAR studies, in accordance with your suggestion.

Authors have not used any standard drug for comparison of Molecular simulation and ADMET studies . I would suggest authors to add that also.

Thank you for your suggestion. We have added the standard drug (Donepezil) for the comparison of molecular simulations and ADMET studies, in accordance with your recommendation.

Comments to the Author

Review #1: The manuscript “Design of novel potent inhibitor based on 2D-QSAR of carbamate derivatives for AChE” caught my attention. The study investigates how the lead molecule M6 interacts with the AChE protein with increased affinity. The results of QSAR and molecular simulations show that the chosen lead molecule could be a good compound for against AChE. Overall, I found the manuscript interesting, but there are many flaws, from the introduction to the conclusion section. A more scientific perspective should guide the rewriting of the entire manuscript. Many paragraphs have more colloquial language.

Thank you for your valuable comments. We have made the necessary corrections throughout the manuscript.

• First, the abstract section mentions the compound as an AChE binder, but further clarification is needed. Has a wet lab already validated this, or is it just an in silico hypothesis? Regarding the docking score, the authors can only give one number instead of three numbers (-11,200 kcal/mol). The abstract does not mention the significance of the results.

Thank you for your feedback. We have taken this into consideration and made the necessary corrections. The clarifications on the compound as an AChE binder, as well as the adjustment of the docking score to a single digit (-11.2 kcal/mol), have been incorporated. In addition, the importance of the results has been highlighted in the abstract.

• In the introduction section:

• . The accumulation of these proteins kills these brain cells [9].---- Here, the authors did not mention any proteins associated with the accumulation.

• The expression “in the last few decades” is not appropriate in this context. Also, it should be mentioned first, second instead of second in the introduction section.

Thank you for your feedback. We have made the necessary changes in the introduction section.

• Analyses pharmacocinétiques prédictives : erreurs

Thank you for your comment. We have corrected it.

typographical• In the section on molecular docking, the authors mentioned that the downloaded crystal structure has a high resolution. However, the given value of 2.35 indicates a lower resolution. The manuscript does not mention the active site residues. Instead, the authors provide the coordinates. Authors should provide the details of the active site of the protein for better understanding.

Thank you for your remark. It is important to note that in crystallography, the lower the resolution value, the better the quality and accuracy of the crystal structure. A resolution of 2.35 Å indicates a sufficiently high quality for detailed structural studies. Therefore, the structure used in this study is of good quality, meeting the requirements for reliable analyses. (A protein with a resolution above 2.7 A˚ is considered to be low-resolution structure, while proteins with a resolution between 2.7 and 1.8 A˚ are classified as medium resolution structures, and those below 1.8 A˚ resolution are typically classified as high-resolution structures [1]).

[1]. BERJANSKII, Mark, et al. Resolution-by-proxy: a simple measure for assessing and comparing the overall quality of NMR protein structures. Journal of biomolecular NMR, 2012, 53: 167-180.‏

Regarding the details on the active site of AChE, we have added them in the manuscript in yellow highlighting for better understanding.

• TIP3P should appear in capital letters in the molecular dynamics section.

• In the molecular docking section “crystallized structure or be used as new leads.” This sentence should be rewritten.

• The search was conducted using AutoDock Vina 4.2 [52]. The sentence should be reworded.

• 6. There are typographical errors, such as residues. Also, Figure 7, located next to Figure 6, should display residues instead of atoms. Authors can label the figures as Fig. 6(a) and Fig. 6(b) instead.

Thank you for your comments. We have taken them into consideration and made the necessary corrections. However, I cannot replace "atoms" with "residues" in this context. For the ligand, the fluctuations are observed at the atom level, since it is a small molecule, and the RMSF is therefore calculated as a function of the atoms. On the other hand, for the protein, which is a macromolecule containing a large number of atoms, it is not relevant to represent the fluctuations for each atom. Therefore, in the protein graph, the RMSF is calculated and represented as a function of the residues. This distinction is essential for a correct interpretation of the results.

• According to the Rg ratio, the values (23.3 and 23) are negligible. The authors stated that their interaction with M6 resulted in a tighter AChE. However, the results given contradict their claims. Here, an increase in the radius of gyration confirms that the protein has become less compact compared to its Apo form. There is no mention of these details in the manuscript.

Based on the radius of gyration (Rg) analysis, the differences between the holo (AChE-M6) and apo (AChE) systems appear to be minimal, with similar values (2.33 Å and 2.3 Å). These differences are not significant and indicate that the interaction with M6 does not substantially modify the compactness of the enzyme. Thus, contrary to the initial assertion, the results suggest that AChE maintains a tight structure even in the presence of M6.

• Hydrogen bonding details: Here, the authors do not make any comparison of the total hydrogen bonds between the apo and holo forms of the AChE protein. The authors focused on the intermolecular hydrogen bond between the ligand and the protein. Instead, they called it intramolecular, which is incorrect.

Thank you for your comment. We performed an analysis of total hydrogen bonds between the apo and holo forms of AChE protein as suggested. Regarding intramolecular hydrogen bonding, we mentioned it in the manuscript as a global information about protein stability, without confusing it with intermolecular hydrogen bonds. Additionally, we added the term intermolecular to clarify and avoid confusion for the reader.

• The manuscript does not mention the necessary residues associated with hydrogen bonds. This will provide an understanding of the consistency in the formation of hydrogen bonds between the protein and the ligand. In addition, the authors did a single run of the simulation instead of triplets. Two more simulations could better validate their results.

Thank you for your remark. We have added a figure showing the residues involved in hydrogen bonding after molecular dynamics, along with a detailed interpretation for better understanding. Additionally, we performed two additional simulations for the complex and the protein to further validate our results, and we found consistent and repeatable outcomes.

• The manuscript should include the MM/PBSA score of the protein-ligand complex for better validation of the protein-ligand complex interaction.

Thank you for your feedback. We performed MM/GBSA analysis for the protein-ligand complex to further validate the interaction, and the results have been included in the manuscript.

• Furthermore, the author made no mention of the structural characteristics of the drug M6 and its functional group• The manuscript can only be considered if the authors revise it according to the suggestions given above.

Thank you for your comment. We have added a detailed description of the structural features of the drug M6 as well as its functional groups in the manuscript, according to your suggestion.

Critique #2: The manuscript is very well written and contains some new findings. There are some minor changes I would like to suggest.

What is meant by M6 in the abstract?

The manuscript is missing some references.

In the Principal Component Analysis section, the term Correlation coefficient is not well described.

Also, what is the coefficient of determination?

Thank you for your review and suggestions. We have clarified in the manuscript what M6 means, and we have added the references in the manuscript, and further detailed the correlation coefficient in the Principal Component Analysis section. In addition, we have included a clear explanation of the coefficient of determination for better understanding.

---

## [Editor Report · Decision Letter 1]

25 Feb 2025

Design of novel potent inhibitor based on 2D-QSAR of carbamate derivatives for AChE inhibition

PONE-D-24-47957R1

Dear Dr. Chtita,

We’re pleased to inform you that your manuscript has been judged scientifically suitable for publication and will be formally accepted for publication once it meets all outstanding technical requirements.

Kind regards,

Sapan Kamleshkumar Shah, Ph.D., M.Pharm.

Academic Editor

PLOS ONE
---

## [Editor Report · Acceptance letter]

PONE-D-24-47957R1

PLOS ONE

Dear Dr. Chtita,

I'm pleased to inform you that your manuscript has been deemed suitable for publication in PLOS ONE. Congratulations! Your manuscript is now being handed over to our production team.

Kind regards,

on behalf of

Dr. Sapan Kamleshkumar Shah

Academic Editor

PLOS ONE